# Angiotensin Type-2 Receptors: Transducers of Natriuresis in the Renal Proximal Tubule

**DOI:** 10.3390/ijms23042317

**Published:** 2022-02-19

**Authors:** Robert M. Carey, Helmy M. Siragy, John J. Gildea, Susanna R. Keller

**Affiliations:** 1Division of Endocrinology and Metabolism, Department of Medicine University of Virginia Health System, Charlottesville, VA 22904, USA; hms7a@hscmail.mcc.virginia.edu (H.M.S.); srk4b@virginia.edu (S.R.K.); 2Department of Pathology, University of Virginia Health System, Charlottesville, VA 22904, USA; jjg5b@virginia.edu

**Keywords:** angiotensin receptor, blood pressure, natriuresis, sodium excretion, renin-angiotensin system, cyclic GMP, protein phosphatase

## Abstract

Angiotensin II (Ang II) type-2 receptors (AT_2_R) are expressed in the adult kidney, prominently in renal proximal tubule cells (RPTCs), and play an important role in opposing renal sodium (Na^+^) retention induced by Ang II stimulation of Ang II type-1 receptor (AT_1_R). Natriuresis induced by AT_1_R blockade is due at least in part to AT_2_R activation and whole body deletion of AT_2_Rs reduces the natriuretic response to increased blood pressure (BP). The major endogenous AT_2_R agonist mediating the natriuretic response is Ang III, the Ang II heptapeptide metabolite generated by aminopeptidase A, and the principal nephron site mediating inhibition of Na^+^ reabsorption by the AT_2_R is the renal proximal tubule (RPT). AT_2_Rs induce natriuresis via a bradykinin, nitric oxide and cyclic GMP (cGMP) signaling cascade. Recent studies demonstrated a key role for protein phosphatase 2A (PP2A) in the AT_2_R-mediated natriuretic response upstream of cGMP. By inducing natriuresis, AT_2_Rs lower BP in the Ang II-infusion model of hypertension. PP2A activation and the natriuretic response to AT_2_R stimulation are defective in spontaneously hypertensive rats, a model of primary hypertension in humans. AT_2_R agonists are candidates for proximal tubule natriuretic agents in Na^+^ and fluid retention disorders.

## 1. Introduction

Primary (essential) hypertension (HT) affects approximately 48% of the US adult population and constitutes a major risk factor for stroke, myocardial infarction, heart failure, and end-stage kidney disease [1,2,3]. Despite this profound threat to public health, research to date has been unable to define the mechanisms that lead to and sustain HT [2]. A current predominant theory for the initiation of HT in both humans and experimental animal models is a fundamental defect in the capacity of the kidneys to excrete sodium (Na^+^) followed by a compensatory increase in renal perfusion pressure to support appropriate Na^+^ excretion. 

The renin-angiotensin system (RAS) is a complex hormonal cascade that is intimately involved in the regulation of renal Na^+^ excretion (U_Na_V) and blood pressure (BP) [4]. The main RAS effector peptide angiotensin II (Ang II) acts at two major receptors, the Ang II type-1 receptor (AT_1_R) inducing renal Na^+^ reabsorption and increasing BP and the Ang II type-2 receptor (AT_2_R) which generally opposes AT_1_R actions. This brief review focuses on the actions and molecular signaling pathways of AT_2_Rs in the induction of natriuresis and lowering of BP in experimental animal models and AT_2_R signaling defects in primary hypertension (summarized in Table 1).

## 2. Renal AT_1_R and AT_2_R Expression

Although both AT_1_Rs and AT_2_Rs are expressed in the adult kidney, AT_1_R mRNA is more abundant than AT_2_R mRNA throughout the nephron and renal microvasculature [4]. However, the AT_2_R typifies several seven-transmembrane G protein-coupled receptors such as dopamine receptors that, despite low levels of mRNA, have significantly higher levels of receptor protein expression in the adult kidney [5]. Reverse transcription PCR and/or immunohistochemistry have demonstrated widespread tubular and vascular distribution of AT_2_R mRNA and protein in the adult kidney, including proximal and distal tubules, collecting ducts, arcuate arteries, afferent arterioles, and outer medullary vasa recta, with dense expression in the vasculature of the renal cortex and the proximal tubules of the outer medulla [6,7]. Subsequent studies have unequivocally confirmed AT_2_R protein in rat renal proximal tubule cells (cytoplasm, mitochondria, and apical plasma membranes) by confocal and electron microscopy immunocytochemistry [8,9,10]. It is important to note that all systematic studies documenting renal tubule expression of AT_2_Rs along the nephron have been conducted in rodent models, and no such studies have yet been reported for the human kidney.

## 3. Evidence for Opposing Actions of AT_2_Rs on AT_1_R-Induced Renal Na^+^ Reabsorption 

AT_2_Rs play a key role in the regulation of renal function, especially Na^+^ and water excretion. The first available evidence for this was provided by AT_2_R-null mice that manifest a combination of pressor and antinatriuretic hypersensitivity to systemic Ang II infusion, wherein the pressure-natriuresis relationship is shifted markedly to the right (less sensitive) in the absence of AT_2_Rs [11]. Pressure-natriuresis is the physiological mechanism whereby an acute rise in renal perfusion pressure (resulting from an increase in BP) triggers an immediate increase in renal Na^+^ excretion which protects against a long-term rise in BP [12,13]. The rightward shift in the pressure-natriuresis curve in AT_2_R-null animals suggested that AT_2_Rs provide a counterregulatory protective role opposing the antinatriuretic actions of Ang II via AT_1_Rs. The fact that AT_2_R-null animals had markedly reduced baseline and Ang II-stimulated renal interstitial fluid bradykinin (BK) and cyclic GMP (cGMP) levels compared with wild-type controls indicated that the response was mediated by BK, nitric oxide (NO) and cGMP [11]. This observation was consistent with earlier findings showing that the BK-NO-cGMP cascade constitutes a major AT_2_R signaling pathway within the kidney [5]. More detailed analyses of the pressure-natriuresis relationship confirmed the rightward shift in AT_2_R-null compared to wild type (WT) animals was independent of a change in glomerular filtration rate, indicating AT_2_R action to reduce renal tubule Na^+^ transport, albeit with a possible contribution from reduced renal blood flow [14,15].

The pressure-natriuresis studies in AT_2_R-null animals suggested the possibility that AT_1_R blockade might induce natriuresis at least in part by activation of AT_2_Rs.Confirming this idea, in a normal (Sprague-Dawley) rat model the natriuretic effect of intrarenal AT_1_R blockade was abolished by concurrent AT_2_R inhibition with selective antagonist PD-123319 (PD) [16]. Nevertheless, because of the predominance of AT_1_Rs over AT_2_Rs in the kidney, it is likely that the natriuretic action of angiotensin receptor blockade is largely due to inhibition of AT_1_Rs.

## 4. Ang III Is the Major Endogenous AT_2_R Agonist for the Natriuretic Response

The endogenous agonist for renal Ang receptors had been assumed to be Ang II, the dominant effector peptide of the RAS. However, the octapeptide was unexpectedly found to be devoid of natriuretic activity at any physiological infusion rate, introducing the possibility that another Ang peptide might serve as the principal endogenous agonist for the natriuretic response [16]. Further studies identified the Ang II metabolite des-aspartyl^1^-Ang II (Ang III) as a potential candidate for the endogenous renal AT_2_R agonist as, in contrast to Ang II, direct intrarenal infusion of Ang III induced a dose-dependent natriuretic response that was abolished by PD [16]. Ang III was subsequently confirmed as the major RAS peptide agonist for AT_2_R-induced natriuresis in a series of experiments employing inhibitors of aminopeptidase A (APA), which converts Ang II to Ang III, and aminopeptidase N (APN), which metabolizes Ang III to smaller inactive peptide fragments (Figure 1) [17,18,19]. Intrarenal infusion of APN inhibitor PC-18 alone induced natriuresis and markedly augmented the natriuretic response to exogenous intrarenal Ang III, and these natriuretic responses were abolished by PD [17,18]. In contrast, when the conversion of Ang II to Ang III was reduced by APA inhibitor EC-33 the natriuretic response to Ang II in the presence of APN inhibition was abolished [18]. Supporting these results, renal interstitial fluid and tissue Ang III levels increased in response to Ang III infusion and were markedly augmented by APN inhibition with PC-18 [19]. In contrast, exogenous Ang (1–7) displayed no natriuretic activity under identical experimental conditions [19]. 

Taken together, the results of in vivo studies confirm that Ang III is the predominant endogenous agonist for renal proximal tubule AT_2_Rs [8,16,17,18,19]. In parallel, in vitro studies have demonstrated that Ang III has significantly enhanced AT_2_R selectivity compared with Ang II, and recent studies indicate that ß-Pro^7^Ang III, an Ang III analog with enhanced AT_2_R selectivity (>20,000-fold greater for AT_2_Rs than AT_1_Rs), also induces natriuresis in the presence of systemic AT_1_R blockade in a normotensive rat model [20,21]. Capitalizing on this knowledge, future studies on the physiology of renal AT_2_Rs could take advantage of β-substituted Ang peptide analogs as peptidomimetics with increased stability to proteolytic degradation and high specificity [22]. In these peptides native α-amino acids within Ang III were replaced by modified β-amino acids to provide unique agonists with enhanced AT_2_R selectivity and prolonged duration of action. 

## 5. Nephron Site of AT_2_R Inhibition of Na^+^ Reabsorption

Although AT_2_Rs are expressed in other nephron segments, the predominant site of AT_2_R action to inhibit renal Na^+^ reabsorption is the proximal tubule. In the two-kidney rat model where the kidney opposite the experimentally infused kidney serves as a vehicle-infused control, intrarenal Ang III or highly selective non-peptide AT_2_R agonist Compound-21 (C-21) infusion increased Na^+^ excretion (U_Na_V), fractional Na^+^ excretion (FE_Na_) and fractional excretion of lithium (FE_Li_) in parallel, indicating that the natriuretic action of AT_2_Rs in vivo is predominantly at the renal proximal tubule [8,19]. No changes in renal hemodynamics (GFR or renal blood flow) were observed in these studies. Furthermore, in vitro real-time imaging of Na^+^ influx using a Na^+^-sensitive fluorescent dye, Na^+^-binding benzofuran isophthalate (SBFI), in cultured rat renal proximal tubule cells (RPTCs) in the presence of AT_1_R blockade with candesartan demonstrated a significant reduction of cellular Na^+^ uptake in response to Ang III but not Ang II (unpublished observations). Finally, electron microscopy immunocytochemistry demonstrated internalization (and inactivation) of the major RPTC Na^+^ transporter Na^+^-H^+^-exchanger-3 (NHE-3) in the RPTC in response to AT_2_R activation [8].

The importance of RPTC AT_2_Rs to induce natriuresis requires validation by selectively deleting these receptors from the RPTC. Whereas studies in the total body AT_2_R knockout model demonstrated defective pressure-natriuresis under a variety of experimental conditions, the definitive role of the RPTC in this response remains to be determined. Key questions to answer include the following. Do selective RPTC AT_2_R-null mice have reduced baseline Na^+^ excretion, and do they have reduced natriuretic responses compared to wild type (WT) mice in response to Na^+^ loading? Do selective RPTC AT_2_R-null mice have defective pressure-natriuresis compared to WT? If so, what cellular mechanisms mediate these responses? Does selective RPTC AT_2_R deletion cause enough Na^+^ retention to increase BP and/or lead to hypertension?

## 6. Renal Mechanism of AT_2_R-Induced Natriuresis

A series of experiments determined the mechanism by which acute AT_2_R activation with C-21 induced inhibition of Na^+^ reabsorption in the RPTC. Confocal and electron microscopy immunocytochemistry and Western blot analysis demonstrated that C-21 acutely stimulated translocation of AT_2_Rs from intracellular sites to the tips of apical plasma membranes (APMs) of RPTCs without altering total cellular AT_2_R expression [8]. APM recruitment is considered an enabling and/or reinforcing mechanism for AT_2_R-induced natriuresis. C-21-induced natriuresis was accompanied by an increase in renal interstitial fluid cGMP levels, and both the cGMP and natriuretic responses were abolished by intrarenal administration of BK B_2_ receptor antagonist icatibant, nitric oxide synthase inhibitor L-NAME, soluble guanylyl cyclase inhibitor ODQ or AT_2_R antagonist PD, supporting a BK-NO-cGMP signaling pathway for AT_2_R-induced natriuresis [8]. The terminal effect on natriuresis was ascribed to internalization and inactivation of major RPTC Na^+^ transporters NHE-3 from apical membranes and Na^+^/K^+^-ATPase (NKA) from basolateral membranes within the RPTC [8]. The final effector in the BK-NO-cGMP signaling pathway is thought to be cGMP. However, whether this cyclic nucleotide acts via stimulation of protein kinase G (PKG) and, if so, by which form of PKG and which downstream pathway(s), needs to be investigated in the future.

## 7. Primary Cell Signaling Pathways Mediating the AT_2_R Natriuretic Response

As stated above, the natriuretic response to AT_2_R activation is largely, if not exclusively, a renal proximal tubule (RPT) event, with a possible small contribution from the thick ascending limb of the loop of Henle [8,23]. AT_2_Rs reduce AT_1_R expression and function via a NO/cGMP/Sp1-dependent mechanism [24]. AT_2_R activation is accompanied by translocation of the AT_2_R to the APM of RPTCs (a mechanism thought to enhance and sustain the natriuretic response) and internalization of major RPTC Na^+^ transporters NHE-3 and Na^+^/K^+^-ATPase (NKA) in a BK/NO/cGMP-dependent manner [8]. Until recently, the major signaling mechanism downstream of AT_2_Rs mediating natriuresis was thought to be the BK-NO-cGMP pathway, which can serve as an intracellular and/or an extracellular signaling pathway. However, for other organs and tissues (e.g., brain, blood vessels) activation of protein phosphatases such as c-Src phosphorylation of Src homology region 2 domain-containing phosphatases (SHP-1), serine threonine protein phosphatase 2A (PP2A) and MAP kinase phosphatase-1 (MKP-1) have been described [4]. Activation of phosphatases inhibits kinase-driven signaling cascades, especially those triggered by AT_1_Rs.

Recent studies strongly suggest that PP2A is a major signaling intermediate downstream of AT_2_Rs and upstream of the cGMP pathway in kidney cells [25,26]. Activating renal AT_2_Rs with C-21 promotes the physical association of AT_2_Rs with PP2A heterotrimer AB55αC, promotes translocation of the PP2A heterotrimer to the APM of RPTCs simultaneously with APM AT_2_R recruitment, and increases renal PP2A activity [26]. As evidence for the functional relevance of PP2A in the AT_2_R natriuretic response, C-21-induced natriuresis, renal cyclic GMP formation, and AT_2_R and PP2A translocation to APMs of RPTCs are all abolished by co-administration of PP2A inhibitor calyculin A [26]. Calyculin A also inhibits other serine-threonine phosphatases, including PP1. Thus, additional more specific approaches will be needed to establish whether the PP2A heterotrimer AB55αC is the major cellular mediator of renal responses to AT_2_R activation. Major cell signaling pathways of renal AT_2_Rs are shown schematically in Figure 2 (left panel).

Although the above-mentioned observations suggest a major role for PP2A AB55αC in AT_2_R signaling, whether this specific PP2A heterotrimer mediates the AT_2_R natriuretic response in RPTCs needs to be validated using, for example, siRNA-mediated knockdown of PP2A subunits or a similar definitive approach. In addition, the precise (intracellular vs apical plasma membrane) site and the extent of physical association of AT_2_Rs with PP2A at baseline and following AT_2_R activation need to be elucidated and molecular characterization of the defect in spontaneously hypertensive rats (SHR; see below) explored. Further, the specific substrate(s) dephosphorylated by PP2A after AT_2_R activation need(s) to be identified. 

## 8. Dopamine-1 Receptor (D_1_R)-AT_2_R Interactions in Natriuresis

In addition to AT_2_Rs, renal dopamine D_1_-like receptors (D_1_Rs) are important natriuretic receptors counterbalancing AT_1_R-mediated tubular Na^+^ reabsorption, and AT_2_R-D_1_R interactions are mutually cooperative and interdependent [5,27] Dopamine-induced natriuresis via D_1_Rs requires the activation of AT_2_Rs and AT_2_R recruitment along microtubules to the APM of RPTCs by an adenylyl cyclase/cAMP/protein kinase C-dependent pathway [28,29]. In addition to D_1_Rs, dopamine D_3_ receptors and AT_2_Rs are synergistic in producing natriuresis and diuresis [30]. Whereas the cooperativity and mutual dependence of D_1_Rs and AT_2_Rs in natriuresis is apparent, the exact roles of each related to changes in whole body Na^+^ status, extracellular fluid volume, and BP changes remain to be defined.

## 9. Sex Determinants of AT_2_R-Induced Natriuresis

Sex as a biological variable has generally been understudied, but it has important implications in cardiovascular/renal disease and hypertension. During their reproductive years women have relatively lower BP and are protected from cardiovascular disease compared to age-matched men, but after menopause BP rises and cardiovascular protection wanes, resulting in, when compared to age-matched men, an increase in the incidence of cardiovascular and renal disease and associated mortality in postmenopausal women [31,32,33].

To explain the age-dependent differences in BP between men and women, the role of AT_2_Rs in renal Na^+^ excretion has been extensively studied in female compared to male rodents [34,35,36]. Renal AT_2_R expression (mRNA) is higher in female than male rats, likely related to the location of the gene encoding the AT_2_R on the X chromosome [4,34]. Consistent with this finding, the pressure-natriuresis relationship was shifted to the left (more sensitive) in female than in male rats, but AT_2_R blockade with PD was found to blunt pressure-natriuresis in both female and male rats to a similar degree [35].

Subsequent studies showed that AT_2_R actions are enhanced when the RAS is activated and that the augmented AT_2_R pathway in females counterbalances the hypertensive effects of Ang II and attenuates the Ang II-dependent resetting of the renal tubuloglomerular feedback (TGF) [36]. The TGF mechanism is a negative feedback control mechanism whereby an increase in flow-dependent electrolyte load to the macula densa region of the distal tubule increases renal afferent arteriolar constriction reducing glomerular filtration rate. The TGF mechanism has been considered important in the regulation of glomerular filtration rate, fluid and electrolyte balance and BP, and Ang II augments the TGF response via AT_1_Rs [37]. An augmented TGF response has been implicated in the pathogenesis of primary hypertension [38].

Extensive comparisons of renal hemodynamic and excretory responses to AT_2_R agonist C-21 in normotensive male and female rats have been reported [8,39]. In one report [39], C-21 induced renal vasodilation and natriuresis in virtually identical fashion in male and female rats and both changes were abolished with PD. These results were consistent with those showing no significant difference between male and female rats in natriuretic responses to C-21 [8]. The renal vasodilatory effect was dose-dependent only in females and was observed to a greater extent in females than males at the highest C-21 doses employed [39]. These studies concluded that AT_2_R activation has similar effects on Na^+^ excretion between the sexes [8,39].

## 10. AT_2_R-Induced BP Reduction in Angiotensin-Dependent Hypertension

The rat Ang II infusion model has been widely employed as an experimental model of human Ang-dependent hypertension. In this model, proximal tubule AT_1_Rs are critical for development of hypertension. Using this model, as anticipated, Ang II markedly increased Na^+^ retention and BP after 24 h of infusion and BP remained elevated for 7 days of continuous infusion. Chronic systemic as well as direct intrarenal infusion of C-21 in Ang II-infused animals abolished Ang II-mediated Na^+^ retention at 24 h, induced a continuously negative Na^+^ balance state compared with Ang II alone during the 7 day experimental period, and reduced BP chronically [40]. Indeed, C-21 was able to reduce BP to control pre-infusion levels when administered either upon initiation of Ang II infusion or at 3 days after hypertension had been established. C-21 infused systemically or intrarenally was effective in the absence of concurrent AT_1_R inhibition [40]. Chronic C-21-induced AT_2_R activation initiated and sustained AT_2_R translocation to RPTC APMs [40]. These results suggest that renal AT_2_R activation can surmount Na^+^ retention under pathological circumstances and may be an appropriate target for pharmacological stimulation of natriuresis and consequent lowering of BP when the RAS is activated.

## 11. Defective AT_2_R-Induced Natriuresis in Primary Hypertension

Spontaneously hypertensive rats (SHR) are inbred rats that develop high BP with increasing age and are widely employed as an Ang II-dependent model of human primary HT. Young pre-hypertensive SHR exhibit increased RPT Na^+^ reabsorption, wherein normal Na^+^ excretion is achieved only at the expense of elevated RPP [41,42,43,44,45,46,47,48,49]. Over time, the kidneys reset to elevated BP in order to continue to excrete a normal Na^+^ load [12,13,49]. Evidence for this kidney-based pathophysiology includes observations that transplantation of pre-hypertensive kidneys from SHR into WKY induces HT in WKY, and that both humans with genetic HT and SHR excrete less Na^+^ and water than controls when RPP is lowered to normotensive levels [50,51,52,53]. In addition, the chronic relationship between arterial pressure and urinary Na^+^ and water output is shifted towards higher BP in SHR compared to WKY, reflecting the adaptation of the kidneys to higher perfusion pressure [54].

AT_2_R-induced natriuresis is defective in both hypertensive and pre-hypertensive SHR [55,56,57]. The defect may account, at least in part, for increased Ang II-dependent Na^+^ reabsorption in SHR. The AT_2_R defect in SHR is at the receptor/post-receptor level and is not due to increased metabolism of endogenous AT_2_R agonist Ang III [56,57]. Neither is the AT_2_R defect in SHR due to reduced AT_2_R expression, as receptor protein levels are equal in SHR and WKY kidneys [26,56,57]. Signaling pathways to stimulate natriuresis involving D_1_R (cAMP/PKA) and AT_2_R (cGMP/PKG) translocation to APMs converge at PP2A [25], and in SHR AT_2_R signaling to PP2A is defective [26]. Impaired AT_2_R and D_1_R signaling in SHR thus likely leads to Na^+^ retention and hypertension by allowing unopposed AT_1_R-mediated renal Na^+^ transport in this animal model of human hypertension. The defects in renal AT_2_R signaling are shown schematically in Figure 2 (right panel).

Although the majority of studies in female SHR have demonstrated an AT_2_R natriuretic defect, one study demonstrated a C-21-induced increase in Na^+^ and water excretion in female but not in male SHR [58]. The other studies [26,55,56,57] consistently show an AT_2_R natriuretic defect in female SHR. The reasons for the differences in results are not clear, but may be related to differences in experimental conditions, use of hypertensive vs prehypertensive SHR, the route of C-21 administration, pretreatment or not with AT_1_R blockade, type of anesthesia and/or differences in data analysis.

Exactly why renal AT_2_Rs actively counterbalance AT_1_R-induced antinatriuresis in the Ang II infusion model of hypertension when the RAS is activated but are defective in SHR, a model of human primary hypertension thought to be related to increased RAS activity, is currently unknown. 

## 12. Enhanced AT_2_R-Induced Natriuresis in Obese Zucker Rats

RPTC AT_2_Rs are especially effective in stimulating natriuresis/diuresis in the obese Zucker rat, a model of obesity, insulin resistance and mild hypertension in humans. AT_2_Rs are upregulated in Zucker rats compared to lean controls and mediate the natriuretic/diuretic effects of AT_1_R blockade [59]. The enhanced natriuresis in this model is accompanied by inhibition of NKA via the NO/cGMP pathway in RPTCs [60]. AT_2_Rs play a protective role against increased BP in obese Zucker rats due to inhibition of proximal tubule Na^+^ reabsorption and the renin suppressive effect of AT_2_Rs [61,62,63,64].

## 13. Potential for AT_2_R Agonists as Natriuretic/Diuretic Agents Acting at the Renal Proximal Tubule

Evidence for functional inhibition of Na^+^ transport at the level of the proximal tubule was provided by showing additive effects of C-21 to those of diuretics acting at the distal tubule (chlorothiazide) or cortical collecting duct (amiloride) [40]. Since approximately 67% of filtered Na^+^ is reabsorbed at the proximal tubule, diuretics acting there would be expected to induce a major increase in Na^+^ load at distal portions of the nephron, the reabsorption of which can be inhibited with existing site-specific natriuretic/diuretic agents. The potential importance of chronic C-21-induced inhibition of Na^+^ reabsorption at the proximal tubule cannot be overstated, as no effective diuretic agents specifically targeting the proximal tubule are currently available. Carbonic anhydrase inhibitors which can induce very mild diuresis/natriuresis at the proximal tubule are not clinically employed as diuretics due to their limited ability to inhibit Na^+^ reabsorption [65]. To validate AT_2_R activation as a viable therapeutic option, it will need to be confirmed that additive or synergistic effects of AT_2_R activation combined with diuretics acting at the distal tubule and/or collecting duct on U_Na_V and free water clearance occur in humans. 

## 14. Summary 

Renal AT_2_Rs acting in tandem with dopamine receptors counterbalance Na^+^ retention elicited by Ang II via AT_1_Rs. The primary cellular mechanisms by which AT_2_Rs induce natriuresis are protein phosphatase PP2A and the BK-NO-cGMP signaling cascade. AT_2_R agonist C-21 can restore Na^+^ balance and reduce BP to baseline in the chronic Ang II infusion model of hypertension. However, AT_2_R-induced natriuresis is defective in SHR. Since AT_2_Rs inhibit Na^+^ transport in the RPT where the majority of Na^+^ is reabsorbed and no effective RPT natriuretic/diuretic agent is currently available or approved for clinical use, AT_2_R agonists appear to be excellent candidates for pharmacologic treatment of Na^+^/fluid retention disorders.

## Figures and Tables

**Figure 1 ijms-23-02317-f001:**
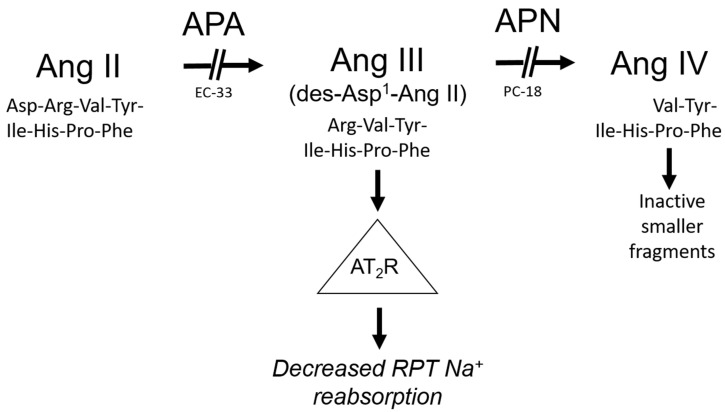
Metabolism of angiotensin II (Ang II) to angiotensin III (Ang III) and smaller inactive peptide fragments by aminopeptidases A (APA) and N (APN), respectively. RPT, renal proximal tubule; EC-33, APA inhibitor; PC-18, APN inhibitor.

**Figure 2 ijms-23-02317-f002:**
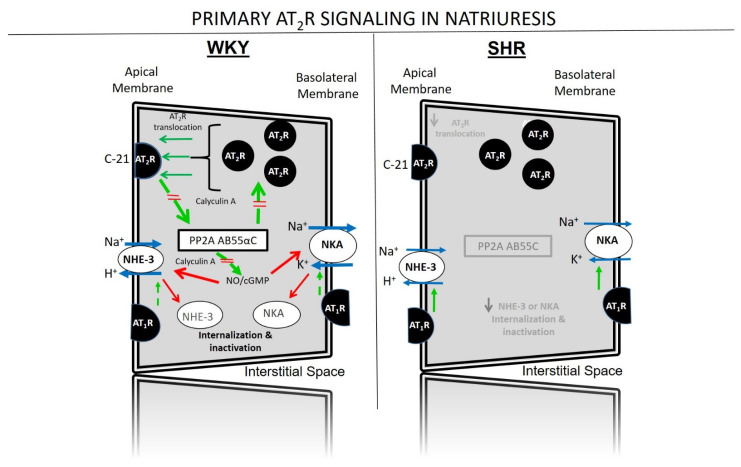
Schematic representation of the angiotensin II (Ang II) type 2 receptor (AT_2_R) protein phosphatase 2A (PP2A) AB55αC signaling pathway by which receptor activation with Compound-21 (C-21) inhibits Na^+^ reabsorption in the renal proximal tubule, increases cyclic GMP (cGMP) production, and induces natriuresis. Green arrows indicate stimulation; red arrows depict internalization and inactivation; red lines depict inhibition; blue arrows stand for effects on ion exchanges by Na^+^ transporters; broken arrows indicate impaired/reduced responses. In Wistar-Kyoto rats (WKY, left), AT2R activation by exogenous non-peptide agonist C-21 stimulates AT2R recruitment from intracellular sites to the apical plasma membranes of renal proximal tubule cells via PP2A-dependent signaling, reinforcing and sustaining the natriuretic response. AT2R activation via a PP2A AB55αC signaling pathway internalizes and inactivates major Na^+^ transporter molecules Na^+^-H^+^-exchanger-3 (NHE-3) and Na^+^-K^+^-ATPase (NKA), counterbalancing angiotensin II (Ang II) type-1 receptor (AT1R) actions to increase Na^+^ reabsorption by stimulating these transporters. In WKY, PP2A inhibition with calyculin A inhibits AT_2_R translocation, cGMP formation, and natriuresis. In spontaneously hypertensive rats (SHR, right) AT2Rs are defective resulting in markedly reduced apical plasma membrane translocation, failure to associate with and activate PP2A, and internalize and inactivate NHE-3 and NKA. Modified from Circ Res. 2022, doi:10.1161/CIRCRESAHA.121.319519 with permission.

**Table 1 ijms-23-02317-t001:** Summary of AT_2_R-mediated Natriuretic Responses in Experimental Animal Models.

Experimental Rodent Model	Natriuretic Response to AT_2_R Agonist	Natriuretic Response Abolished by AT_2_R Antagonist	Natriuretic Response Abolished by Inhibition of NO/cGMP	Natriuretic Response Abolished by Calyculin A (PP2A Inhibitor)
**Sprague-Dawley rat**				
Normal baseline	Yes	Yes	Yes	-
AT_1_R blockade (acute)	-	Yes	-	-
Ang II infusion (chronic)	Yes	Yes	-	-
**Wistar-Kyoto rat**	Yes	Yes	Yes	Yes
**Spontaneously hypertensive rat**				
Prehypertensive	No	NA	NA	-
Hypertensive	No	NA	NA	-
**Zucker rat**				
Obese	Yes	Yes	Yes	-
Lean	No	NA	NA	-

Comparison of natriuretic responses to modifications of AT_2_R signaling among experimental animal models. NA, non-applicable; -, experiment not been carried out; AT_1_R, angiotensin type-1 receptor; AT_2_R, angiotensin type-2 receptor; NO, nitric oxide; cGMP, cyclic GMP; PP2A, protein phosphatase 2A.

## Data Availability

Not applicable.

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
