# Peer review of "Angiotensin Type-2 Receptors: Transducers of Natriuresis in the Renal Proximal Tubule"

_ijms, 2022, doi:10.3390/ijms23042317_

Round 1
Reviewer 1 Report
General Comments:
This is an excellent overview manuscript written by a group of well-recognized investigators and leaders in the renin-angiotensin system in the kidney, with an emphasis on the kidney angiotensin II (Ang II) type 2 (AT2R) receptors and its roles in the natriuretic response in renal physiology and hypertension. Most investigators have extensively studied the roles and mechanisms of Ang II type 1 (AT1R) receptor-mediated effects in the kidney and hypertension, whereas those mediated by Ang II AT2Rs remain less studied and less well understood. Against this background, the excellent contributions by Carey et al. in this manuscript comprehensively reviews and updates the advances in this particular field, which is very timely and necessary. Overall, the authors systemically review and discuss the key findings from their group reported over many years using different animal models with AT2R-specific agonists or antagonists infused either systemically or via intra-cortex interstitial infusion. The data are impressive and largely convincing and it is expected that these studies will provide new insights into the important roles of Ang II or Ang III via acting on AT2R in the regulation of proximal tubule sodium reabsorption, natriuretic responses, and blood pressure. The manuscript is well written and I don't have many comments, but the authors are encouraged to consider the following comments to revise their manuscript.
Specific Comments:
1. The manuscript has been written very well and concisely to the points throughout. However, I don't mind if the authors may consider to expand a little bit in each of 12 or 13 subsections by adding a few sentences discussing the current status of the research in each of these particular topics. Then they proceed to discuss major discoveries from their group or other groups in the field.
2. By addressing the above comments, the authors would also be able to expand the reference list a bit by including and discussing other investigators' studies in addition the authors' in the field.
3. Abstract: Please use "angiotensin II" to angiotensin type-2 receptor, as other angiotensin receptors may not bind and activate type 2 receptors.
3. Abstract: Please use "angiotensin II" to angiotensin type-1 receptors, which is more accurate for AT1R.
4. Abstract: the statement that "Natriuresis induced by AT1R blockade is due to AT2R activation" may be revised as "in part due to", as that statement is probably too strong. Indeed, this is because most likely that AT1R antagonists act primarily to block AT1R.
4. Introduction: Line 38: The statement "the angiotensin type-2 receptor (AT2R) which generally opposes AT2R actions" It should read "generally opposes AT1R actions".
5. Lines 79-80: The statement that "The pressure-natriuresis studies in AT2R-null animals suggested the possibility that AT1R blockade might induce natriuresis by activation of AT2Rs". I would prefer to add "at least in part", otherwise it implies " primarily" or "exclusively"?
Thank you.
Author Response
Comment: This is an excellent overview manuscript written by a group of well-recognized investigators and leaders in the renin-angiotensin system in the kidney with an emphasis on kidney angiotensin II type 2 receptors and their role in the natriuretic response in renal physiology and hypertension.
Response: We thank Reviewer 1 for these comments and for the excellent feedback.
Comment: The manuscript has been written very well and concisely to the points throughout. However, I don’t mind if the authors may consider to expand a little bit in each of the 12 or 13 subsections by adding a few sentences discussing the current status of the research in each of these particular topics. Then they proceed to discuss major discoveries from their group or other groups in the field.
Response: We thank the reviewer for this comment. We have added a few sentences for some but not all of the subtopics as appropriate.
Comment: By addressing the above comments, the authors would also be able to expand the reference list a bit by including and discussing other investigators’ studies in addition to the authors’ in the field.
Response: As suggested, we have added a few references for these comments when appropriate.
Comment: Abstract: Please use “angiotensin II” to angiotensin type-2 receptor, as other angiotensin receptors may not bind and activate type 2 receptors.
Response: We have changed “angiotensin type-2 receptor” to “angiotensin II (Ang II) type 2 receptor” as suggested.
Comment: Abstract: The statement that “Natriuresis induced by AT1R blockade is due to AT2R activation” may be revised as “in part due to”, as that statement is probably too strong. Indeed, this is because most likely that AT1R antagonists act primarily to block AT1R.
Response: We agree and have completed the suggested change.
Comment: Introduction, Line 38: The statement “the angiotensin type-2 receptor (AT2R) which generally opposes AT2R actions” should read “generally opposes AT1R actions”.
Response: Thank you for picking up this typo, which is now corrected.
Comment: Lines 79-80: The statement that “The pressure-natriuresis studies in AT2R-null animals suggested the possibility that AT1R blockade might induce natriuresis by activation of AT2Rs”, I would prefer to add “at least in part”; otherwise it implies “primarily” or “exclusively”.
Response: We agree and made the change.
Reviewer 2 Report
The author showed that angiotensin type-2 receptors (AT2R) are expressed in the adult kidney, prominently in renal proximal tubule cells (RPTCs), and play an important role in opposing renal sodium (Na+) retention induced by angiotensin (Ang) II stimulation of the angiotensin type-1 receptor (AT1R). They discussed that the role for protein phosphatase 2A (PP2A) in the AT2R-mediated natriuretic response upstream of cGMP. By inducing natriuresis, AT2Rs lower BP in the Ang II-infusion model of hypertension. The subject matter of this review is laudable and of interest to the renin-angiotensin community. I have some minor questions.
1) Are angiotensin type-2 receptors (AT2R) expressed in the adult kidney, prominently in renal proximal tubule cells (RPTCs) in human?
2) The author described that stimulation of renal AT2Rs promotes the physical association of AT2Rs with PP2A heterotrimer AB55αC, promotes translocation of the PP2A heterotrimer to the APM of RPTCs simultaneously with APM AT2R recruitment, and increases renal PP2A activity.23 As evidence for the functional relevance of PP2A in the AT2R natriuretic response, C-21-induced natriuresis, renal cyclic GMP formation, and AT2R and PP2A translocation to APMs of RPTCs are all abolished by co-administration of PP2A inhibitor calyculin A. Do these mechanisms work in humans? If you have such knowledge, please add it.
3) Please state your thoughts or evidence on the importance of renal AT2R in renal proximal tubule cells in the development of essential hypertension and the treatment of hypertension.
4) Please indicate the keyword (line 23)
Author Response
Comment: The subject matter of the review is laudable and of interest to the renin-angiotensin community.
Response: We thank Reviewer 2 for these comments and excellent feedback.
Comment: Are angiotensin type-2 receptors expressed in the adult kidney, prominently in renal proximal tubule cells in human?
Response: Yes, as shown in references 22 and 24 (Gildea JJ et al.), AT2Rs are expressed in cultured human renal proximal tubule cells. However, to our knowledge, a detailed study of the renal distribution of AT2Rs in the human kidney has not been conducted, so the comparative distribution of AT2Rs in different epithelial cell types along the nephron is unavailable.
Comment: The author described that stimulation of renal AT2Rs promotes the physical association of AT2Rs with the PP2A heterotrimer AB55αC, promotes translocation of the PP2A heterotrimer to the APM of RPTCs simultaneously with APM AT2R recruitment, and increases renal PP2A activity. As evidence for the functional relevance of PP2A in the AT2R natriuretic response, C-21-induced natriuresis, renal cyclic GMP formation and AT2R and PP2A translocation to APMS of RPTCs are all abolished by co-administration of PP2A inhibitor calyculin A. Do these mechanisms work in humans? If you have such knowledge, please add it.
Response: Reference 23 (Kemp et al.) is the first study reporting these mechanisms (in rats). To our knowledge, no studies on these mechanisms have been performed for human renal proximal tubule cells.
Comment: Please state your thoughts or evidence on the importance of renal AT2R in renal proximal tubule cells in the development of essential hypertension and the treatment of hypertension.
Response: As indicated in the manuscript, when the renin-angiotensin system is activated (Ang II infusion model), AT2Rs oppose the sodium-retaining actions of AT1Rs and lower BP. However, in spontaneously hypertensive rats, a renal proximal tubule AT2R signaling defect exists wherein the sodium retaining actions of AT1Rs are unopposed. We currently do not know whether and how this defect contributes to the development of hypertension or what the basis for this signaling defect is, but the defect is not due to a reduction in the baseline expression of AT2Rs in the proximal tubule.
This manuscript is a resubmission of an earlier submission. The following is a list of the peer review reports and author responses from that submission.